# Recurrent Convolutional Neural Networks for 3D Mandible Segmentation in Computed Tomography

**DOI:** 10.3390/jpm11060492

**Published:** 2021-05-31

**Authors:** Bingjiang Qiu, Jiapan Guo, Joep Kraeima, Haye Hendrik Glas, Weichuan Zhang, Ronald J. H. Borra, Max Johannes Hendrikus Witjes, Peter M. A. van Ooijen

**Affiliations:** 13D Lab, University Medical Center Groningen, University of Groningen, Hanzeplein 1, 9713GZ Groningen, The Netherlands; b.qiu@umcg.nl (B.Q.); j.kraeima@umcg.nl (J.K.); h.h.glas@umcg.nl (H.H.G.); m.j.h.witjes@umcg.nl (M.J.H.W.); 2Department of Radiation Oncology, University Medical Center Groningen, University of Groningen, Hanzeplein 1, 9713GZ Groningen, The Netherlands; p.m.a.van.ooijen@umcg.nl; 3Data Science Center in Health (DASH), University Medical Center Groningen, University of Groningen, Hanzeplein 1, 9713GZ Groningen, The Netherlands; 4Department of Oral and Maxillofacial Surgery, University Medical Center Groningen, University of Groningen, Hanzeplein 1, 9713GZ Groningen, The Netherlands; 5Institute for Integrated and Intelligent System, Griffith University, Nathan, QLD 4111, Australia; zwc2003@163.com; 6CSIRO Data61, Epping, NSW 1710, Australia; 7Medical Imaging Center (MIC), University Medical Center Groningen, University of Groningen, Hanzeplein 1, 9713GZ Groningen, The Netherlands; r.j.h.borra@umcg.nl

**Keywords:** accurate mandible segmentation, oral and maxillofacial surgery, 3D virtual surgical planning (3D VSP), convolutional neural network

## Abstract

Purpose: Classic encoder–decoder-based convolutional neural network (EDCNN) approaches cannot accurately segment detailed anatomical structures of the mandible in computed tomography (CT), for instance, condyles and coronoids of the mandible, which are often affected by noise and metal artifacts. The main reason is that EDCNN approaches ignore the anatomical connectivity of the organs. In this paper, we propose a novel CNN-based 3D mandible segmentation approach that has the ability to accurately segment detailed anatomical structures. Methods: Different from the classic EDCNNs that need to slice or crop the whole CT scan into 2D slices or 3D patches during the segmentation process, our proposed approach can perform mandible segmentation on complete 3D CT scans. The proposed method, namely, RCNNSeg, adopts the structure of the recurrent neural networks to form a directed acyclic graph in order to enable recurrent connections between adjacent nodes to retain their connectivity. Each node then functions as a classic EDCNN to segment a single slice in the CT scan. Our proposed approach can perform 3D mandible segmentation on sequential data of any varied lengths and does not require a large computation cost. The proposed RCNNSeg was evaluated on 109 head and neck CT scans from a local dataset and 40 scans from the PDDCA public dataset. The final accuracy of the proposed RCNNSeg was evaluated by calculating the Dice similarity coefficient (DSC), average symmetric surface distance (ASD), and 95% Hausdorff distance (95HD) between the reference standard and the automated segmentation. Results: The proposed RCNNSeg outperforms the EDCNN-based approaches on both datasets and yields superior quantitative and qualitative performances when compared to the state-of-the-art approaches on the PDDCA dataset. The proposed RCNNSeg generated the most accurate segmentations with an average DSC of 97.48%, ASD of 0.2170 mm, and 95HD of 2.6562 mm on 109 CT scans, and an average DSC of 95.10%, ASD of 0.1367 mm, and 95HD of 1.3560 mm on the PDDCA dataset. Conclusions: The proposed RCNNSeg method generated more accurate automated segmentations than those of the other classic EDCNN segmentation techniques in terms of quantitative and qualitative evaluation. The proposed RCNNSeg has potential for automatic mandible segmentation by learning spatially structured information.

## 1. Introduction

Oral cancer is a type of cancer that originates from the lip, mouth, or upper throat [1]. Globally, there are an estimated 354,864 new cases of oral cancer annually, and there were 177,384 deaths in 2018 [2]. Surgical tumor resection is the most common curative treatment for oral cancer [3]. During surgical removal of malignant tumors in the oral cavity, a continuous resection of the jaw bone can be required. Currently, this resection is based on the 3D virtual surgical planning (VSP) [4,5] that enables accurate planning of the resection margin around the tumor, taking into account the surrounding jaw bone. Research [5,6] has indicated that 3D VSP requires accurate delineation of mandible organs, which is manually performed by technologists. However, manual mandible delineation in CT scans is very time consuming (about 40 min) and has high inter-rater variabilities (Dice score of 94.09% between two clinical experts) [7], and the performances of technologists can also be affected by fatigue [7,8,9]. In order to help improve the reliability and efficiency of the manual delineation, robust and accurate algorithms for automatic mandible segmentation are highly demanded for the 3D VSP [7,10].

In general, the existing mandible segmentation approaches can be divided into two categories [11], i.e., traditional and deep learning-based approaches. Traditional approaches [12,13,14,15,16,17] have been widely investigated for automatic mandible segmentation in CT scans. Gollmer et al. [12] proposed a fully automatic segmentation approach that uses a statistical shape model for mandible segmentation in cone-beam CT. Chen et al. [13] presented an automatic multi-atlas model that registered CT slices with the obtained atlas to enable multi-organ segmentation in head and neck CT scans. Mannion-Haworth et al. [14] proposed automatic active appearance models (AAMs) that rely on a group-wise registration method to generate high-quality anatomical correspondences. Albrecht et al. [15] combined multi-atlas segmentation and the active shape model (ASM) for the segmentation of automatic organs at risk (OARs) in head and neck CT scans. Torosdagli et al. [16] proposed an automatic two-step strategy that first uses random forest regression to localize the mandible region and then performs the 3D gradient-based fuzzy connectedness algorithm for mandible delineation. The aforementioned approaches offer potentially time-saving solutions. However, these methods mostly rely on the averaged shapes or atlases of the structures generated by domain experts [12,13,14,15]. Therefore, traditional approaches lead to poor individualization for single cases in mandible segmentation [18,19,20].

Alternatively, deep learning approaches have demonstrated a strong ability in automatic descriptive and detailed image feature extraction [21,22]. Ibragimov et al. [23] applied a tri-planar patch-based CNN for mandible segmentation in head and neck CT scans and achieved superior performance for most of the OAR segmentation when compared to the existing approaches. Zhu et al. [24] applied a 3D Unet that uses a loss function combining Dice score and focal loss for the training of the network. This 3D Unet-based method achieved better performances than that of the state-of-the-art approaches in OAR segmentation. Tong et al. [25] proposed a fully CNN (FCNN) with a shape representation model for mandible segmentation in CT scans and achieved better results than those of the conventional approaches. Egger et al. [26] implemented a three-step training strategy of the fully convectional networks proposed by [27] to segment the mandibles in the CT scans on a locally acquired dataset. Qiu et al. [11] presented a 2D Unet-based mandible segmentation approach that segments mandibles on orthogonal planes of the CT scans. Liang et al. [28] proposed a multi-view spatial aggregation frame for joint localization and the segmentation of the OARs and achieved competitive segmentation performance.

These aforementioned deep learning approaches for mandible segmentation have the encoder–decoder-based CNN architectures that consist of an encoder and a decoder. The encoder network maps a given input image to a feature space that is then processed by a decoder to produce an output image of the same size as the input [29]. Ye et al. [30] investigated the geometry of the EDCNN and demonstrated that its excellent performances come from the expressiveness of the symmetric encoder and decoder networks and the skip connections that enable feature flows from the encoder to the decoder.

The use of deep learning approaches helped to achieve better performance than the conventional approaches. However, our research indicates that challenges still exist in EDCNN-based approaches for mandible segmentation in CT scans. On the one hand, they are often affected by the noise or metal artifacts that are commonly present in head and neck CT scans [31,32]. On the other hand, they are not robust in segmenting organs that have weak boundaries, such as the condyles and coronoids of the mandible [11]. The main reason is that the inputs of EDCNN-based segmentation approaches are either truncated into 2D slices or 3D patches due to the computation capacity [33]. These truncated inputs do not represent the complete anatomical structures of the organs and, as a consequence, lead to inaccurate segmentation of the detailed structures and artifact corrupted regions [33].

In this paper, we proposed a novel CNN-based 3D mandible segmentation approach named recurrent convolutional neural networks for mandible segmentation (RCNNSeg). RCNNSeg has the ability to accurately segment the detailed anatomical structures. Different from the classic EDCNN approaches that need to slice or truncate the CT scan into 2D slices or 3D patches during the segmentation process [33], the RCNNSeg approach can perform mandible segmentation on complete 3D CT scans. The proposed method adopts the structure of the recurrent neural networks that form a directed acyclic graph to enable recurrent connections between adjacent nodes to retain their connectivity. Each node then functions as a classic EDCNN to segment a single slice in the CT scan. The proposed segmentation structure enables one to identify the shape of the structures based on their anatomical connectivity. Our approach can perform 3D mandible segmentation on sequential data of any varied length and does not require large computation cost. Chen et al. [34] and Bai et al. [35] presented similar ideas that adopt a long short-term memory (LSTM) recurrent neural network [36] and a fully convolutional network (FCN) for image segmentation. However, FCN and convolutional LSTM were trained separately due to the huge demand of computing resources from the convolutional LSTM [34,35]. Moreover, the method from [35] allows a batch size of 1 in the training of the LSTM. In order to solve the issue of lacking computing resources, we chose the vanilla RNN instead of LSTM, which allows the whole network to be trained in an end-to-end manner, while LSTM needs to be trained in the decoupled manner, as demonstrated in [34,35].

Our major contributions are three-fold. First, we proposed a novel mandible segmentation architecture that uses recurrent neural networks to facilitate connected units between adjacent slices in an anatomical sequence of scanning. Each unit is then implemented as a classic encoder–decoder segmentation architecture for 2D slice segmentation. Second, the state-of-the-art 3D segmentation algorithms [37,38] usually down-sample the raw data to overcome memory issues. In contrast, the implementation of the RCNNSeg approach enables 3D segmentation in a way that reduces computational complexity of the model without loss of image quality. The 3D mandible segmentation can be performed for any varied length of the scan. Third, the proposed approach is able to tackle the problems of truncated inputs, which represent the main reason for the underperformance on detailed structures and regions that are corrupted by metal artifacts in CT scans. Extensive experiments were performed on two head and neck CT scans, and the proposed RCNNseg approach outperformed the existing segmentation approaches for automatic mandible segmentation.

The remainder of the paper is organized as follows: Section 2 introduces our proposed RCNNSeg approach for automatic mandible segmentation. In Section 3, extensive experiments on two head and neck CT datasets are presented. The proposed method is compared both qualitatively and quantitatively with the EDCNN-based approaches as well as state-of-the-art methods for mandible segmentation. In Section 4, different aspects of the proposed method are discussed. Conclusions are given in Section 5.

## 2. Materials and Methods

Due to extensive demands on computational resources, the encoder–decoder-based mandible segmentation methods, such as Unet [29], SegNet [39], and AttUnet [40], cannot directly deal with the 3D medical images. Instead, three strategies are widely used for the feeding of 3D volumetric data, which are shown in Figure 1. The first two strategies in Figure 1a,b use either 2D slices or a few adjacent slices as inputs to feed into the EDCNNs, which are truncated along the z-axis from the volumetric data. The third one in Figure 1c uses 3D patches as inputs, which are cropped along all three orthogonal axes. These strategies are easy to implement and can deal with the large memory demands from deep learning algorithms; however, they cannot take into account the complete anatomical structures of the organs in CT scans that often frustrate the accurate segmentation of detailed structures.

Therefore, we propose a novel mandible segmentation approach, which adopts the structure of the recurrent neural network to build successive nodes that enable connectivity between adjacent nodes. Meanwhile, each node then functions as a classic 2D EDCNN to achieve mandible segmentation on a single CT slice.

### 2.1. Feedforward, Feedback, and Recurrent Networks

There are three main neuron connections within the visual cortex, namely feedforward, recurrent synapses, and feedback connections [41]. The feedforward (feedback) connections usually bring inputs (outputs) to a later (earlier) stage of the cortex region along (the opposite direction of) the processing pathway [42], which are widely implemented in most of the deep learning architectures [21,43], including the classic EDCNNs [29,39,40]. Recurrent synapses, however, usually exceed the feedforward and feedback connections and interconnect neurons at the same stage of the pathway in the cortex [41,42]. Due to the interconnection mechanism of the recurrent neural networks, they are more applicable to tasks based on processing of sequential data.

### 2.2. Recurrent Convolutional Neural Networks for Mandible Segmentation (RCNNSeg)

We proposed a recurrent convolutional neural networks for mandible segmentation in order to accurately segment mandibles in the head and neck CT scans. Different from the recurrent convolutional neural networks for object recognition [42,44] that build recurrent connections among the nodes at the same layer, the proposed RCNNSeg enables recurrent connections between adjacent successive units, each of which is then formed as a 2D segmentation network to process a single slice of the scan.

Our proposed architecture for 3D image segmentation can be observed in Figure 2a. The RCNNSeg takes a sequence of CT slices as inputs and then outputs a sequence of the corresponding segmented images of the same length. The RCNNSeg architecture can be unfolded to the structure shown in Figure 2b. RCNNSeg consists of varied lengths of successive units to be able to process sequential data of different lengths. Each of the units has a structure of the classic 2D EDCNN to evoke slice-based mandible segmentation.

At the *t*th unit, the 2D EDCNN segmentation unit takes as input the *t*th CT slice $I_{t}$ concatenated with the output $O_{t-1}$ from the previous time step t−1. Then, the output of this unit is given as
(1)$$O_{t}=w_{f}^{⊤}*\mathrm{concat}\left(I_{t},w_{r}^{⊤}*O_{t-1}\right),$$
where ⊤ is the transpose. $w_{f}$ and $w_{r}$ denote the feedforward weights in the 2D segmentation and the recurrent weights between the adjacent units, respectively.

The 2D segmentation unit adopts the classic EDCNN-based mandible segmentation approaches, such as U-Net [29], SegU-Net [39], and Att U-Net [40]. During the training, the feedforward weights $w_{f}$ from the EDCNN and the recurrent weights $w_{r}$ are shared with all the units at different time steps.

**Loss function** Given an input scan, we optimize the feedforward weights with the gradient-based optimization technique. The cost function at unit *t* is a combination of the Dice loss [45] and the binary cross entropy (BCE) loss [46], which is given as
(2)$$L^{t}=\alpha \times L_{\mathrm{BCE}}^{t}+\beta \times L_{\mathrm{Dice}}^{t},$$
where α and β control the amount of contributions the BCE and Dice terms give in the loss function L, respectively. For a detailed explanation and implementation, we refer the interested readers to [45,46]. It is worth noting that these two loss functions have the potential to deal with imbalanced data, which fit well with our case.

**Training of the model** For the training of the RCNNSeg, we use the backpropogation through time (BPTT) [47] technique that begins by unfolding a recurrent neural network into successive units. As shown in Figure 2, the unfolded network contains *N* units, each of which takes a slice as the input in the ordered sequence and outputs the corresponding segmentation result. According to Equation (Equation 2), the loss at the *t*th step is denoted as $L^{t}$. The total loss for a given ordered sequence of slices is the sum of the loss over all of the steps. We then optimize all of the parameters of the model based on the chain rule [48] to minimize the total loss.

### 2.3. Experimental Setup

#### 2.3.1. Implementation Details

Our proposed approach was implemented in Pytorch [49]. All of our experiments were performed on a workstation equipped with Nvidia K40 GPUs with 12 GB of memory. We set the weights of the loss function to 0.5 for both the BCE loss term α and the Dice loss term β. We used Adam optimization [50] with a learning rate of $r=10^{-4}$. The total number of epochs was 40 and 80 for the UMCG dataset and the PDDCA dataset, respectively. Moreover, an early stopping strategy was utilized if there was no improvement in the loss of the validation set for 10 epochs in order to avoid over-fitting.

#### 2.3.2. Evaluation Metrics

For quantitative analysis of the experimental results, several performance metrics are considered, including the Dice similarity coefficient (DSC), average symmetric surface distance (ASD), and 95% Hausdorff distance (95HD).

The Dice similarity coefficient (DSC) is often used to measure the consistency between two objects [51]. Therefore, it is widely applied as a metric to evaluate the performance of image segmentation algorithms. We do not elaborate further about the DSC and refer the readers to [51].

The average symmetric surface distance (ASD) [25] computes the average distance between the boundaries of two object regions. It is used to measure the error between the surfaces of the ground truth and the segmented regions. It is defined as
(3)$$D_{\mathrm{ASD}}\left(A,B\right)=\frac{d(A,B)+d(B,A)}{2},$$
(4)$$d\left(A,B\right)=\frac{1}{N}\sum_{a\in A}\min_{b\in B}∥a-b∥,$$
where ∥.∥ is the $L_{2}$ norm. a and b are the coordinates of the points on the boundary of objects *A* and *B*, respectively.

Hausdorff distance (HD) measures the maximum distance of a point in a set *A* to the nearest point in the other set *B*. It is defined as
(5)$$D_{\mathrm{HD}}\left(A,B\right)=\max\left(h\left(A,B\right),h\left(B,A\right)\right),$$
(6)$$h\left(A,B\right)=\max_{a\in A}\min_{b\in B}∥a-b∥,$$
where h(A,B) is often called the directed HD. The maximum HD is sensitive to contours. When the image is contaminated by noise or occlusion, the original Hausdorff distance is prone to mismatch [52,53]. Huttenlocher et al. [54] proposed the concept of partial Hausdorff distance. The 95HD is similar to maximum HD and selects 95% of the closest points in set *B* to the point in set *A* in Equation (Equation 6) to calculate h(A,B)
(7)$$D_{95\mathrm{HD}}=\max\left(h^{95\%}\left(A,B\right),h^{95\%}\left(B,A\right)\right),$$
(8)$$h^{95\%}\left(A,B\right)=\max_{a\in A}\min_{b\in B^{95\%}}∥a-b∥.$$

The purpose of using this metric is to eliminate the impact of a very small subset of inaccurate segmentation on the evaluation of the overall segmentation performance.

## 3. Results

In this section, we demonstrate the effectiveness of the proposed approach for mandible segmentation in CT scans. We verify this by both quantitative and qualitative experimental results on two datasets, namely, the local UMCG dataset and the public PDDCA dataset [52], and facilitate comparisons with the state-of-the-art methods.

### 3.1. The UMCG Head and Neck Dataset

The UMCG head and neck dataset was collected in the department of oral and maxillofacial surgery at the University Medical Center Groningen. This dataset contains 109 head and neck CT scans reconstructed with a kernel of Siemens Br64, I70 h(s) or B70s. Each scan consists of 221 to 955 slices with a size of 512×512 pixels. The pixel spacing varies from 0.35 to 0.66 mm, and the slice thickness varies from 0.6 to 0.75 mm. The corresponding manual mandible segmentations were obtained by an experienced researcher using Mimics software version 20.0 (Materialise, Leuven, Belgium) and then confirmed by a clinician.

We compare our proposed RCNNSeg approach with the classic EDCNN approaches for mandible segmentation. For brevity, we refer to our methods as RUnetSeg, RAttUnetSeg, and RSegUnetSeg, which use U-Net [29], Att U-Net [40], and SegU-Net [39] as the base units of the proposed RCNNSeg, respectively. We also investigated the EDCNN approaches with three different strategies for the feeding of the 3D volumetric data. The three strategies are in the 2D, 2.5D, and 3D forms that take 2D slices, three consecutive 2D slices, and 3D patches as inputs, respectively. We randomly chose 90 cases as training, 2 cases as validation, and 17 cases as test models. The models were trained from scratch, and the training took approximately 40 h, while the test on one scan took about 1.5 min.

**Quantitative Results** We evaluated the proposed RCNNSeg for mandible segmentation based on the metrics mentioned in Section 2.3.2. In Table 1, we list the results on DSC, $D_{\mathrm{ASD}}$, and $D_{95\mathrm{HD}}$ as well as the corresponding standard deviations. In general, our proposed RCNNSeg-based approaches outperformed the EDCNN-based methods on mandible segmentation. For the DSC, all RCNNSeg-based approaches performed better than the corresponding EDCNNs with different strategies, with an improvement of at least 0.79%. Our proposed RCNNSeg also achieved the minimum error on $D_{\mathrm{ASD}}$. On $D_{95\mathrm{HD}}$, except for RUnetSeg, which underperformed the 2.5D U-Net, all other RCNNSeg-based approaches outperformed the corresponding EDCNNs. It is also surprising that the 3D patch EDCNNs dramatically underperformed when compared with all the other approaches.

**Qualitative results** We illustrate the 3D view of an example taken from the UMCG head and neck dataset in Figure 3. Compared to the ground truth in Figure 3a, the 2D and 2.5D EDCNN-based segmentation approaches shown in Figure 3b,c,f,g,j,k failed to segment the mandible detailed structures, such as the coronoids (indicated by red circles) and parts of the mandible body (indicated by the yellow circles). The 3D patch EDCNN approaches shown in Figure 3d,h,l managed to acquire most of the mandible segmented, but they also segmented part of the skull as mandible by mistake. The results of our proposed RCNNSeg approaches in Figure 3e,f,m show more accurate segmentation of those detailed structures with much less segmentation on other bone structures.

Figure 4 illustrates examples of three slices in the aforementioned encircled regions in a 2D slice view. The three images in Figure 4a are the corresponding ground truth (GT) segmentations. The images shown in Figure 4b,f,j (the middle column) are results from the 2D EDCNNs. The third and fourth columns in Figure 4 show the results from 2.5D EDCNNS and 3D patch EDCNNs. The last column gives the results from the proposed RCNNSeg approaches. Pink indicates the regions that are missed by the corresponding segmentation approaches. In general, the 2D and 2.5D EDCNN approaches missed more regions than the RCNNSeg approaches in segmenting the detailed structures. The 3D patch-based EDCNNs seem to segment the mandible quite well, but they also segment other bone structures, such as the skull, as mandibles.

### 3.2. PDDCA Dataset

We also evaluated the proposed pipeline on the public dataset PDDCA [52]. This dataset contains 48 patient CT scans from the Radiation Therapy Oncology Group (RTOG) 0522 study with manual segmentation of the left and right parotid glands, brainstem, optic chiasm, and mandible. Each scan consists of 76 to 360 slices with a size of 512×512 pixels. The pixel spacing varies from 0.76 to 1.27 mm, and the slice thickness varies from 1.25 to 3.0 mm. We followed the same training and testing split as described in [52]. Forty out of the 48 patients in PDDCA with manual mandible annotations were used in previous studies [52,55], in which the dataset was split into training and test subsets with 25 (0522c0001–0522c0328) and 15 (0522c0555–0522c0878) cases, respectively [52]. We used the pre-trained models obtained from the UMCG dataset and fine-tuned them on the training subset of the PDDCA dataset. We evaluated the performances of the models on the test subset.

**Quantitative results**Table 2 shows the quantitative evaluation based on the average DSC, ASD, and 95HD used in the challenge [25,52]. The achieved performance indicates that our proposed RCNNSeg-based methods outperformed the corresponding EDCNN-based approaches with different input feeding strategies regarding all evaluation metrics.

In Table 3, we also compare our proposed approach with the state-of-the-art methods on the PDDCA dataset. We mark in bold the best three performances for each evaluation metric. In general, our RCNNSeg-based approaches outperformed the other existing approaches. Only the RAttUnetSeg model underperformed when compared with some of the existing approaches [56,57,58] on DSC and 95HD.

## 4. Discussion

Over the last few decades, many traditional computer vision algorithms [13,14,15] and CNN-based methods [23,24,25,28,56,57,58,59] have been proposed for head and neck CT segmentation. However, there are still significant challenges to completely automate the segmentation of the mandible from CT scans, whereas manual delineation is time consuming and has high inter-rater variabilities [7]. The quantitative comparison in Table 3 shows that the CNN algorithms outperform the traditional SSM- or atlas-based methods in general. Therefore, the development of CNN algorithms can bring significant benefits to automatically segment the mandible. However, these methods are proposed for OAR segmentation in radiotherapy rather than 3D VSP.

In this paper, we present a robust end-to-end deep learning approach for accurate mandible segmentation in 3D VSP. The proposed RCNNSeg approach adopts the structure of the recurrent convolutional neural networks to enable connections between adjacent slices in the CT volume. Unlike the classic encoder–decoder-based approaches that need to truncate the 3D volume, our approach can perform 3D mandible segmentation on sequential data of any varied length and does not require large computational resources.

Quantitative and qualitative evaluation on two datasets demonstrates that our proposed approach is robust in mandible segmentation, as illustrated in Figure 3 and Figure 4 and Table 1, Table 2 and Table 3. Comparisons with the classic EDCNN approaches and state-of-the-art approaches illustrate that the RCNNseg approach significantly improves segmentation of the mandible. It is worth noting that RCNNSeg is very robust for the segmentation of the weak and blurry boundaries (for instance, the coronoids of mandibles).

The experimental results show that the proposed RCNNSeg is feasible and effective for 3D mandible segmentation in CT scans. The RCNNSeg approach can also be applied to other segmentation tasks that need to deal with 3D sequential data. The architecture of the RCNNSeg adopts the structure of the recurrent neural network that facilitates recurrent connections between adjacent slices. The proposed approach implemented the strategy that takes advantage of the anatomical prior information from the previous slice and then segments the current CT slice.

This strategy takes advantage of shape prior information, which considers the segmentation result of the previous slice as the shape prior to the segmentation of the current slice. Therefore, the proposed approach can help to identify the continuous structure of the mandible in 2D segmentation networks. In addition, the proposed approach utilizes the RNN module that helps the extraction of spatial information of the object based on the collection of context and shape information. This strategy can support further research on 3D image segmentation. It can also help alleviate memory issues for 3D medical image segmentation as well as segmentation tasks on sequential data, such as video images.

In this study, we used the 109 H&N CT dataset and a small public dataset to validate our method, which cannot satisfactorily represent the average patient population that requires tumor resection surgery. In practical automatic segmentation simulation, larger amounts of data across regions should be further explored. Moreover, orthognathic surgery and complex trauma have been widely reported in using 3D VSP [60]. Most of the CT scans in the datasets exclude metal implants or dental braces that often lead to noisy and blurry structures and also make the segmentation tasks difficult. Validation on even more scans with dental braces or metal implants can be performed. In addition, automatic mandible segmentation from MRI images may be required in the future, since MRI-based 3D surgery planning workflow has been developed [61]. In our future research, further validation and evaluation will be performed to determine whether our proposed approach is effective for real clinical practice.

## 5. Conclusions

We propose an end-to-end approach for the accurate segmentation of the mandible from H&N CT scans. Our approach incorporates the encoder–decoder-based segmentation algorithms into recurrent connections and uses a combination of Dice and BCE as the loss function. We implemented the proposed approach on 109 H&N CT scans from our dataset and 40 scans from the PDDCA public dataset. The experimental results show that the RSegCNN strategy can yield more significant performance than that of the conventional algorithms in terms of quantitative and qualitative evaluation. The proposed RSegCNN has potential for automatic mandible segmentation by learning spatial structured information.

## Figures and Tables

**Figure 1 jpm-11-00492-f001:**
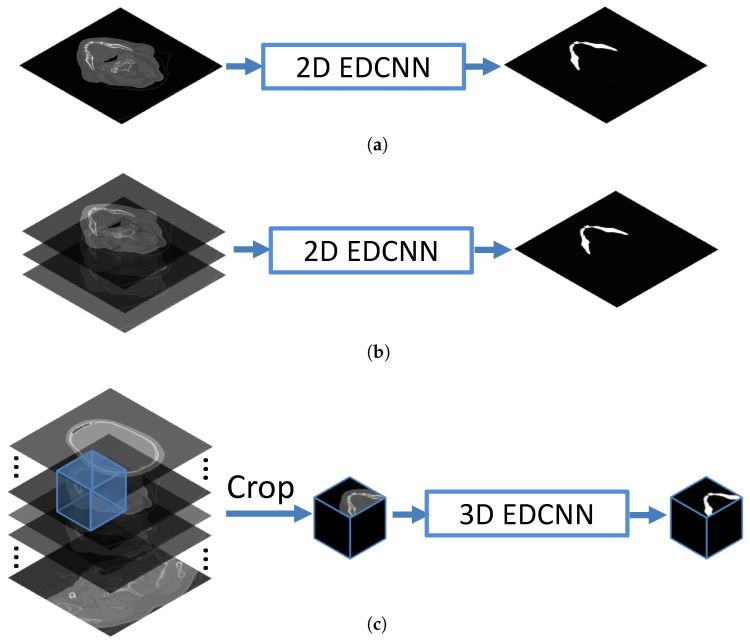
Illustration of three prevalent strategies for the feeding of input data to the classic encoder–decoder-based convolutional neural networks. (**a**) Use of 2D EDCNN network for 2D slice-based object segmentation. (**b**) Use of 2D EDCNN for the image segmentation based on several adjacent slices from the volumetric data. (**c**) Use of 3D EDCNN based on 3D patches cropped from the complete volumetric data.

**Figure 2 jpm-11-00492-f002:**
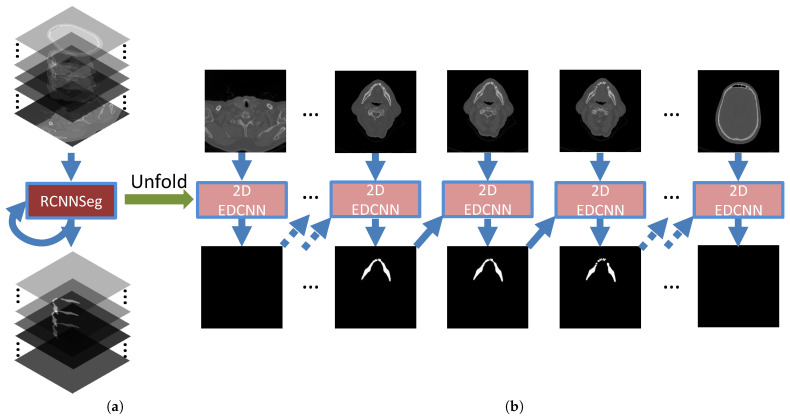
The overall graphic scheme of the proposed methods. The architecture of RCNNSeg with two components: (**a**) the RCNNSeg and its loss drawn with recurrent connections; (**b**) the same seen as a time-unfolded computational graph, where each node is now associated with one particular time instance.

**Figure 3 jpm-11-00492-f003:**
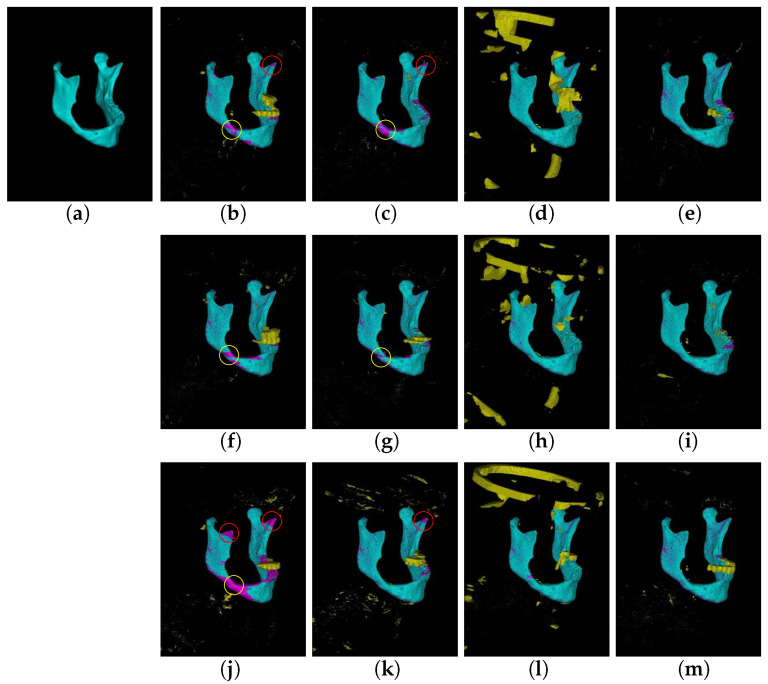
3D view of a case from the UMCG head and neck dataset. (**a**–**m**) Ground truth, 2D U-Net, 2.5D U-Net, 3D U-Net, RUnetSeg, 2D SegU-Net, 2.5D SegU-Net, 3D U-Net, RSegUnetSeg, 2D Att U-Net, 2.5D Att U-Net, 3D Att U-Net, and RAttUnetSeg. We use cyan to indicate the correctly segmented mandible compared to the ground truth. Pink represents the regions that were missed by the algorithms, while yellow indicates the segmented regions that are not the mandible. The red circles indicate the coronoids of the mandibles that are often missed by the traditional EDCNNs, and the yellow circles indicate parts of the mandible body.

**Figure 4 jpm-11-00492-f004:**
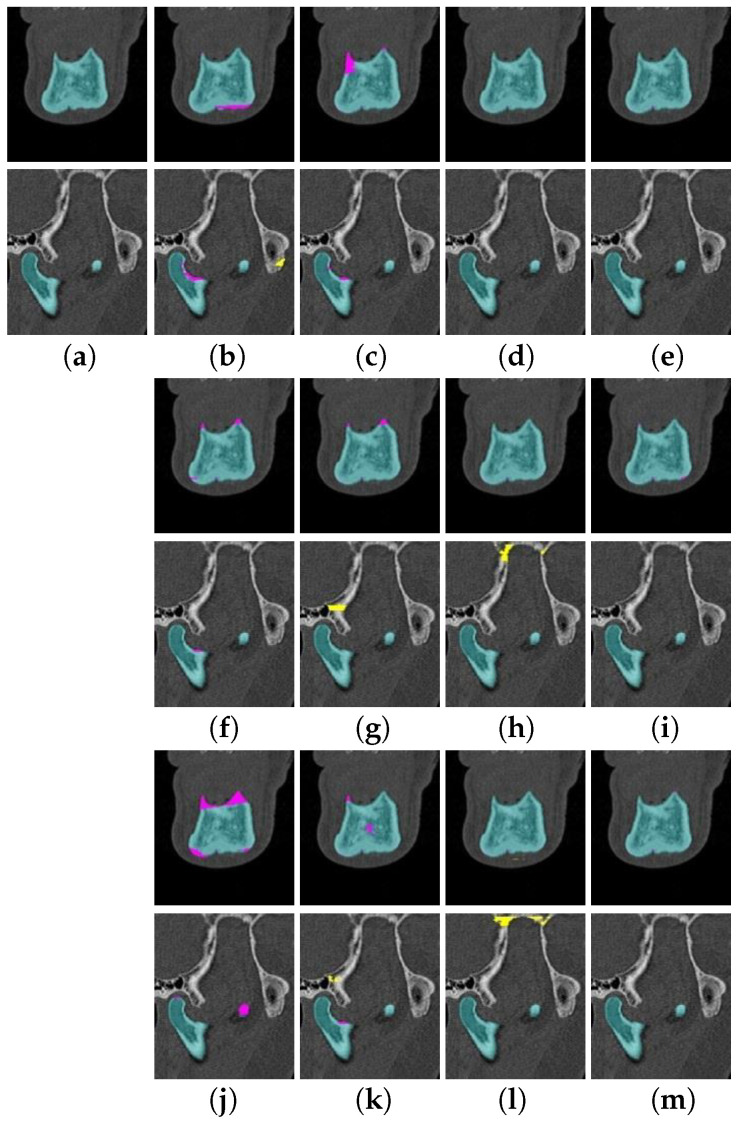
Examples of the automatic segmentation of mandibles in the UMCG dataset. (**a**) Ground truth segmentation on the three examples. (**b**–**m**) Segmentation results obtained on the example slices from 2D U-Net, 2.5D U-Net, 3D U-Net, RUnetSeg, 2D SegU-Net, 2.5D SegU-Net, 3D SegU-Net, RSegUnetSeg, 2D Att U-Net, 2.5D Att U-Net, 3D Att U-Net, and RAttUnetSeg. Cyan indicates the correctly segmented mandible compared to the ground truth. Pink represents the regions that were missed by the algorithms, while yellow indicates the segmented non-mandible regions.

**Table 1 jpm-11-00492-t001:** Quantitative comparison of segmentation performance in the UMCG dataset between the proposed RCNNSeg and the classic EDCNNs. The values in the square brackets indicate the standard deviation of the corresponding measurements. We mark in bold the best performance in each metric.

	DSC (%)	$D_{ASD}$ (mm)	$D_{95HD}$ (mm)
2D U-Net	95.95 [±2.24]	0.3615 [±0.3366]	4.0145 [±4.6487]
2.5D U-Net	96.34 [±1.99]	0.4053 [±0.7565]	**2.0154 [±1.5949]**
3D U-Net	77.73 [±6.74]	17.2808 [±6.0045]	133.7464 [±22.8779]
RUnetSeg	**97.53 [±1.65]**	**0.2070 [±0.2623]**	2.3975 [±4.6051]
2D SegU-Net	96.30 [±2.06]	0.2794 [±0.2447]	3.7958 [±4.3662]
2.5D SegU-Net	96.69 [±2.12]	0.4210 [±0.6111]	4.9574 [±7.5637]
3D SegU-Net	81.88 [±7.14]	19.0109 [±6.7765]	137.2283 [±17.1781]
RSegUnetSeg	**97.48 [±1.70]**	**0.2170 [±0.3491]**	**2.6562 [±5.7014]**
2D Att U-Net	94.21 [±3.34]	0.6929 [±0.8370]	5.1368 [±3.2194]
2.5D Att U-Net	93.87 [±2.89]	0.5188 [±0.3327]	4.9223 [±4.6204]
3D Att U-Net	83.92 [±5.43]	16.2428 [±3.9300]	124.1773 [±14.2461]
RAttUnetSeg	**96.57 [±1.69]**	**0.2978 [±0.2340]**	**2.4068 [±1.5479]**

**Table 2 jpm-11-00492-t002:** Quantitative comparison of the segmentation performance between the proposed RCNNSeg-based approaches and the EDCNN-based methods on the PDDCA dataset. The values in the square brackets indicate the standard deviation of the corresponding measurements. We mark in bold the best performance in each metric.

	DSC (%)	$D_{ASD}$ (mm)	$D_{95HD}$ (mm)
2D U-Net	94.15 [±1.31]	0.1827 [±0.0915]	2.0547 [±1.4431]
2.5D U-Net	94.19 [±1.25]	0.1915 [±0.0669]	1.7512 [±0.6539]
3D U-Net	91.85 [±5.32]	3.7577 [±6.1869]	36.9138 [±66.9059]
RUnetSeg	**94.71 [±1.35]**	**0.1353 [±0.0614]**	**1.4098 [±0.8573]**
2D SegU-Net	94.69 [±1.33]	0.1765 [±0.0671]	1.5067 [±0.6938]
2.5D SegU-Net	94.76 [±1.20]	0.1532 [±0.0622]	1.6856 [±0.6426]
3D SegU-Net	93.08 [±2.80]	2.4289 [±5.8637]	24.1133 [±62.1808]
RSegUnetSeg	**95.10 [±1.21]**	**0.1367 [±0.0382]**	**1.356 [±0.4487]**
2D Att U-Net	92.99 [±1.25]	0.2924 [±0.2523]	3.1848 [±4.0571]
2.5D Att U-Net	92.75 [±1.34]	0.2502 [±0.0887]	2.1815 [±1.0656]
3D Att U-Net	90.14 [±8.50]	6.3894 [±11.7528]	54.2182 [±72.3141]
RAttUnetSeg	**93.87 [±1.29]**	**0.1773 [±0.0515]**	**1.6397 [±0.6219]**

**Table 3 jpm-11-00492-t003:** Comparison of segmentation performance between the state-of-the-art methods and our proposed RCNNSeg approach; bold font indicates the best three performers for each measurement.

	DSC (%)	$D_{ASD}$ (mm)	$D_{95HD}$ (mm)
Multi-atlas [13]	91.7 [±2.34]	-	2.4887 [±0.7610]
AAM [14]	92.67 [±1]	-	1.9767 [±0.5945]
ASM [15]	88.13 [±5.55]	-	2.832 [±1.1772]
CNN [23]	89.5 [±3.6]	-	-
NLGM [59]	93.08 [±2.36]	-	-
AnatomyNet [24]	92.51 [±2]	-	6.28 [±2.21]
FCNN [25]	92.07 [±1.15]	0.51 [±0.12]	2.01 [±0.83]
FCNN+SRM [25]	93.6 [±1.21]	0.371 [±0.11]	1.5 [±0.32]
CNN+BD [57]	**94.6 [±0.7]**	0.29 [±0.03]	-
HVR [56]	94.4 [± 1.3]	0.43 [± 0.12]	-
Cascade 3D U-Net [58]	93 [±1.9]	-	**1.26 [±0.5]**
Multi-view [28]	94.1 [±0.7]	0.28 [±0.14]	-
RUnetSeg	**94.71 [±1.35]**	**0.1353 [±0.0614]**	**1.4098 [±0.8573]**
RSegUnetSeg	**95.10 [±1.21]**	**0.1367 [±0.0382]**	**1.3560 [±0.4487]**
RAttUnetSeg	93.87 [±1.29]	**0.1773 [±0.0515]**	1.6397 [±0.6219]

## Data Availability

The Public Domain Database for Computational Anatomy Dataset (PDDCA) is available at https://www.imagenglab.com/newsite/pddca/ (accessed on 24 January 2019). Unfortunately, for reasons of ethics and patient confidentiality, we are not able to provide the sequencing data in a public database. The data underlying the results presented in the study are available from the corresponding author.

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
