# Peer review of "Recurrent Convolutional Neural Networks for 3D Mandible Segmentation in Computed Tomography"

_jpm, 2021, doi:10.3390/jpm11060492_

Round 1

Reviewer 1 Report

The topic covered in this article is very interesting. It was carried out with scientific rigor and the conclusions are consistent with the results.

Reviewer 2 Report

The manuscript „Recurrent Convolutional Neural Networks for 3D Mandible Segmentation in Computed Tomography” is very interesting and well written.

However, for the clinicians it is very hard to follow and requires a deep knowledge of the theoretical aspects of a computer tomography.

For dental and maxillofacial region, Cone Beam Computed Tomography (CBCT) is more adequate, involving less ionizing radiation exposure with good accuracy and complying with ALARA principles.

So, in my opinion, despite of the scientific value, the manuscript is not adequate, in the present form, to be published in Journal of Personalized Medicine.

Reviewer 3 Report

This is a sound and useful paper that presents a novel approach for producing detailed anatomical structures in computed tomography.  as such, subject to three minor corrections, the paper merits publication more or less as submitted.

The necessary changes are:

Line 125: Replace "In contrary..." with "In contrast..."

Line 306: Not "... we presented..." but "... we present..."

Line 313: Change the verb to singular, i.e. "are" to "is".

Once these changes are made the manuscript should be accepted.

Reviewer 4 Report

Interesting study.

Round 2

Reviewer 2 Report

The authors addressed the reviewers' concerns.